# Gene Expression Analysis of Microtubers of Potato *Solanum tuberosum* L. Induced in Cytokinin Containing Medium and Osmotic Stress

**DOI:** 10.3390/plants10050876

**Published:** 2021-04-27

**Authors:** Lisset Herrera-Isidron, Eliana Valencia-Lozano, Pablo Yamild Rosiles-Loeza, Maria Guadalupe Robles-Hernández, Abigail Napsuciale-Heredia, Jose Luis Cabrera-Ponce

**Affiliations:** 1Unidad Profesional Interdisciplinaria de Ingeniería Campus Guanajuato (UPIIG), Instituto Politécnico Nacional, Av. Mineral de Valenciana 200, Puerto Interior, Silao de la Victoria, Guanajuato 36275, Mexico; lissetupiig@gmail.com (L.H.-I.); pablo.rosiles@cinvestav.mx (P.Y.R.-L.); lupitaroblesh316@gmail.com (M.G.R.-H.); a.napsuciale@gmail.com (A.N.-H.); 2Departamento de Ingeniería Genética, Centro de Investigación y de Estudios Avanzados del IPN, Unidad Irapuato, Irapuato 36824, Mexico; eliana.valencia@cinvestav.mx

**Keywords:** microtuberization, two component system, cytokinin signaling, *Solanum tuberosum*, potato

## Abstract

Potato microtuber productions through in vitro techniques are ideal propagules for producing high quality seed potatoes. Microtuber development is influenced by several factors, i.e., high content sucrose and cytokinins are among them. To understand a molecular mechanism of microtuberization using osmotic stress and cytokinin signaling will help us to elucidate this process. We demonstrate in this work a rapid and efficient protocol for microtuber development and gene expression analysis. Medium with high content of sucrose and gelrite supplemented with 2iP as cytokinin under darkness condition produced the higher quantity and quality of microtubers. Gene expression analysis of genes involved in the two-component signaling system (*StHK1*), cytokinin signaling, (*StHK3*, *StHP4*, *StRR1*) homeodomains (*WUSCHEL*, *POTH1*, *BEL5*), auxin signaling, *ARF5*, carbon metabolism (*TPI*, *TIM)*, protein synthesis, *NAC5* and a morphogenetic regulator of tuberization (*POTH15*) was performed by qPCR real time. Differential gene expression was observed during microtuber development. Gene regulation of two component and cytokinin signaling is taking place during this developmental process, yielding more microtubers. Further analysis of each component is required to elucidate it.

## 1. Introduction

Potato (*Solanum tuberosum* L.) is the fourth most important crop worldwide, with an average production of 388 million tons per year of tubers [1,2]. A potato tuber is a specialized stem that arises from the underground organ known as stolon [3]. Tubers are used for plant survival by vegetative propagation; they are sink organs in which surplus photosynthetic assimilates are stored [4], with starch [5,6], vitamins [7] and proteins [8,9] as the main storage components.

Plant biotechnology by means of in vitro tissue culture has been applied to produce potato tubers, called microtubers [10], minitubers [11,12] and vitrotubers [13].

Microtuber development has advantages provided by the handling, production and packaging of healthy seeds. The technique accelerates the multiplication process, producing more seed potato faster and cheaper than other methods. Researchers estimate that growers can earn 40% more from apical cuttings than from minitubers [14].

The induction of microtubers under in vitro conditions was first described by Baker [15]; Mes and Menge [16], using high content of sucrose in the medium, as plant pathology tool. An extensive number of articles since then have been published for basic and practical research [17,18,19,20].

The most critical factor for tuber development in potato plants is attributed to sucrose (SUC) [17]. SUC has been demonstrated to be essential for its osmotic effect [21], as an energy source, at higher content as signaling molecule [22], for review see. [17]. In microtuber development, growth rates depend on SUC availability [21] using radiolabeled sugars, and demonstrated that more sugar is translocated to microtubers when SUC, rather than glucose or fructose, is the carbon source. They concluded that SUC acts primarily as suitable carbon source for uptake and utilization by the plantlets, but, at 8%, it also provides a favorable osmolarity for the development of microtubers. Microtuber growth rate depends on SUC availability since SUC hydrolisis to glucose and fructose limited it [23]. SUC enhance tuber development, although it is unlikely that SUC itself is the tuber-inducing stimulus. It is considered that tuber development is induced rather by balance between inhibitory (i.e., gibberellins) and promoting plant growth regulators.

An extensive number of publications about potato microtuber induction using plant growth regulators have been made: auxins [24], gibberellins [25,26,27,28,29], strigolactones [30], abscisic acid [31], ethylene [32], jasmonate [3,33,34,35,36].

Cytokinins (CKs) are plant growth regulators (plant hormones) that promote potato tuber development; vascularity and root induction of potatoes is the effect of cytokinins (CKs) [37,38,39,40,41,42,43].

Furthermore, overexpression of *POTH1* and *StBEL5* (transcriptional factors, that regulate tuberization by targeting genes that control growth) increases cytokinin levels at the tips of stolons and promotes tuberization [44].

*StBEL5* induces the activity of three *LOG* genes of potato, which encode for enzymes involved in cytokinin activation. Overexpression of *LOG1* in tomato (*Solanum lycopersicum*) generated tubers from axillary buds [45], suggesting that this enzyme is a critical component of the tuberization pathway.

In addition, CKs participate in plant defense against biotic and abiotic adverse factors [46,47,48]. All the above indicates the important role of CKs in both the formation of tubers and the general development and resistance of potato plants [49,50,51,52,53].

CKs’ perception and signal transduction in *Arabidopsis thaliana* involves a His-Asp phosphorelay (MSP) similar to that found in bacterial two component signaling systems (TCS), major routes to sense and respond environmental stimuli. TCS has been implicated in control of the stem cell pool of the shoot apical meristem (SAM), leaf and root differentiation (root apical meristem, RAM), vasculature patterning, chloroplast biogenesis, photomorphogenesis, apical dominance, gravitropism, fertility, seed development, senescence and stress tolerance [54,55,56,57,58,59,60].

Cheng et al. [61], proposed from a physiological and proteomic analysis, a molecular mechanism model of metabolic pathways activated by exogenous cytokinins during tuber development (metabolism, bioenergy, storage, redox homeostasis, cell defense, rescue, transcription, translation, chaperones, signaling, and transport-related proteins).

Lomin et al. [52] published a detailed analysis of all basic elements of CK signal transduction in potato (*Phureja* genome). They found that potato contains all molecular elements for CK signaling via multistep phosphorelay (MSP), multiple alleles of histidine kinase genes (*StHK*), typical CK-binding, with *StHK* exhibiting organ-specific expression pattern and promoter activity hardly affected by CKs. Authors pointed out that peculiarities in CK perception apparatus might be associated with tuber development.

In this study, we used a dynamic network to propose a minimal model of the osmotic-protein regulatory network that integrates the previously reported cross-talk between the CK signaling pathways, with transcriptional regulators that have been shown to be important in tuber development. We described microtuber formation in potato *S. tuberosum* cv. Alpha, under osmotic stress in the presence of cytokinins, and analyzed gene expression by qPCR of the coding genes involved in the two-component signaling system (*StHK1*), cytokinin signaling, (*StHK3*, *StHP4*, *StRR1*) homeodomains (*WUSCHEL*, *POTH1*, *BEL5*), auxin signaling, *ARF5*, carbon metabolism (*TPI*, *TIM*), protein synthesis, *NAC5* and a morphogenetic regulator of tuberization (*POTH15*).

## 2. Results

### 2.1. Microtuber Induction

Microtuberization was induced from explants derived from in vitro propagated shoots of potato cv. Alpha in solid osmotic stress medium, MR8-G6-2iP (medium MS [62], sucrose 8%, gelrite 6 g/L, 2iP 10 mg/L, −1.5 Mpa) after one month of incubation in darkness condition (Figure 1A). Microtubers were 7 mm in diameter, 2.3 per explant and 126.3 mg fresh weight. Explants cultured in non-osmotic medium developed only small swellings in the tip of shoot explants (Figure 1B).

### 2.2. Interaction Analysis of Proteins Directly Involved in Microtuberization

To understand a possible regulatory interaction, network of a STRING-based bioinformatic analysis with confidence (0.500) was made based on potato *S. tuberosum* genome (Figure 2).

A set of two component and cytokinin signaling-related coding genes were evaluated (*StHK1, StHK3, StHP4* and *StRR1*) and according with the theoretical gene network in *S. tuberosum*, those proteins interact with *WUSCHEL*, *POTH1* and *BEL5* homeodomains and with morphogenic genes, such as *POTH1*5, *ARF5* (MP), carbon metabolism (*TIM, TPI*) and protein synthesis (*NAC5*) whose expression underwent significant changes during the tuberization of *S. tuberosum*.

Interestingly, *StHK1* and *POTH15* were pivotal in this network interaction, suggesting a greater importance for those proteins in the theoretical regulatory network in the tuberization of *S. tuberosum*.

This interaction network represents the first report of theoretical and preliminary evidence in the microtuberization process involving the two component and cytokinin signaling coupled with *STBEL5*, *POTH1*.

### 2.3. Gene Expression Analysis during Microtuberization

Genes involved in two component and cytokinin signaling, homeodomains, auxin signaling, master regulators of tuberization and carbon metabolism, were analyzed in non-osmotic conditions MR1-G3-2iP (MS medium supplemented with 1% sucrose, gelrite 3 g/L) and osmotic stress (MR8-G6-2iP) (MS medium supplemented with 8% sucrose, gelrite 6 g/L); both media with presence of 2iP (N6-(2-Isopentenyl adenine) 10 mg/L as cytokinin.

Expression levels of *StHK1* (histidine kinase 1), that functions as an osmosensor that detects water stress, salinity, control stomatal, and it is related to signaling cascade of MAP Kinases (MAPK), was upregulated (Log2) in MR8-G6-2iP (osmotic stress) having as control MR1-G3-2iP (non-osmotic stress) medium, 5.44 at 8 days, 3.30 at 15 days and downregulated −1.56 at 23 days and −0.18 at 31 days (Figure 3, Table 1).

*StHK3* (histidine kinase 3) is a negative regulator of the adaptive response to osmotic stress; it was downregulated in MR8-G6-2iP (osmotic stress) medium −2.32 to 8 days, −0.43 to 15 days, −1.62 to 23 days, 0.71 to 31 days (Figure 3, Table 1).

Levels of expression of the histidine-containing phosphotransferase, *AHP4*, was downregulated in the 8 and 31 days, expression level to −3.78 and −0.83 respectively in 15 to 23 days, expression level was 0.53 and 0.43 (Figure 3, Table 1).

A response regulator type B, *StRR1* was downregulated −1.34 to 8 days and −1.77 to 23 days in MR8-G6-2iP (osmotic stress) and upregulated 0.14 to 15 days and 3.79 in 31 days (Figure 3, Table 1).

*WUSCHEL* and *POTH1*, related homeobox were upregulated in MR8-G6-2iP (osmotic stress), in three times, 8, 15, 31 dayss, and downregulated in 23 days; this suppresses the differentiation, allows the maturation of the tubers and prevents the emergence of shoots in the newly formed tuber; expression level 2.76, 3.31, −0.48, 0.74 and 0.28, 0.87, −3.02, 0.98 (Figure 3, Table 1).

*POTH15*, orthologue of *STM* (shoot meristem less), was found downregulated −0.83 at 8 days, −1.47 at 23 days and upregulated 2.10 at 15 days and 1.34 at 31 days (Figure 3, Table 1).

*BEL5*, bell-like homeodomain protein 5, was downregulated −1.39 at 8 days, −0.48 at 23 days and upregulated 0.64 at 15 dayss and 1.35 at 31 days (Figure 3, Table 1).

*NAC5*, nascent chain-associated complex is a highly conserved protein complex, and was upregulated 2.08 at 15 dayss, 0.89 at 23 dayss and downregulated −0.41 at 8 days, −0.89 at 31 days (Figure 3, Table 1).

*TIM* and *TPI*, triosephosphate isomerase, were expressly opposed. *TIM* was upregulated in three first times, 1.11, 2.58, 3.51, and downregulated in 31 days −4.17. TPI was downregulated in three first times, −0.44, −0.37, −1.78 and upregulated 0.88 at 31 days (Figure 3, Table 1).

*ARF5*, an auxin response factor 5, was downregulated −0.32 at 8 days, −1.08 at 23 days and upregulated 1.22 at 15 days and 31 days in osmotic stress medium (Figure 3, Table 1).

## 3. Discussion

Potato tuber formation is a complex developmental process that requires the interaction of environmental, biochemical and genetic factors. It involves many important biological processes, including carbon partitioning, signal transduction and meristem determination [47,48,49]. Under optimal crop field conditions, tuberization in potato is activated by signals that function in the leaf and move down into stolon tips to induce and activate tuber formation. The major signals that regulate the onset of tuber formation in potato are: CYCLING DOF FACTOR (*StCDF1*), *StBEL5* and SELF-PRUNING6A (*StSP6A*) as mobile signals originating in the leaf [63,64,65].

In plant tissue culture medium with 2–3% sucrose lacking plant growth regulators, microtubers can be induced after 4–5 months in culture. However, microtubers are significantly accelerated and improved by using plant growth regulators or by changing culture conditions.

The most efficient factors for microtuber induction are increased sucrose concentrations (5–8%) [15,37,38,39,40,41,42,43] and the addition of cytokinins. Wakasa et al. [20] analyzed gene expression using microtubers and were found to be an excellent tool for the analysis of potato genes that are expressed in normal tubers. In our protocol, potato microtuberization was achieved in osmotic stress conditions using sucrose (8%) and high content of gelrite (6 g/L) compared to non-osmotic stress (MS, sucrose 1%, gelrite 3 g/L) MR1-G3-2iP after four weeks of incubation.

Gelrite is a bacterial (*Pseudomonas elodea*) polysaccharide composed of glucuronic acid, rhamnose and glucose, and has been used routinely to promote somatic embryo maturation of several plant species with concentrations exceeding the standard 3 g/L up to 12 g/L. It has been reported that high gel strength was associated with reduced water availability from the medium to the explants [66]. Gelrite influences cytokinin-sensitivity in the moss *Physcomitrella* patens inducing a bud protonema differentiation in the mutant cytokinin-sensitive *PC22* [67]. The gelrite effect is attributable to physical and chemical properties of the gelling agent [68]. We found a positive effect on number, size and germination of microtubers induced in gelrite 6 g/L, compared in non-osmotic conditions. Our results demonstrate a synergism between gelrite concentration and cytokinin signaling, in addition to other genes strongly involved in tuber formation. Furthermore, the cytokinin 2iP was the best to induce microtuber development in our protocol. Lomin et al. [52] found that the highest affinity with potato histidine kinases was 2iP and trans-zeatin, compared with cis-zeatin, BAP, kinetin and thidiazuron.

In the present work, we propose a model of the osmotic-protein regulatory network that integrates the previously reported cross-talk between the CK signaling pathways, with transcriptional regulators that have been shown to be important in tuber development (Figure 4).

TCS is considered as one of the most crucial signal transduction systems in plants. Evidence suggests that TCS pathways are involved in sensing the environmental stimuli, ethylene signaling, light perception, circadian rhythm and cytokinin-dependent processes which include shoot and root development, vascular differentiation and leaf senescence [54,55,69]. Cytokinin signaling has been associated with the variety of stress response [70]. Histidine kinase of the TCS is known to function as an oxidative stress sensor [71], involved in the primary cell wall [72]. The genome of potato *S. tuberosum* contains 4 genes encoding histidine kinases (HK), 7 genes involved in phosphotransferase (HPT), 8 B-type response regulators (B-ARR), 8 A-type response regulators (A-ARR) and 4 C-type response regulators (C-ARR) [52].

These genes were devised in STRING database and are involved in the two-component signaling system (*StHK1*), cytokinin signaling, (*StHK3*, *StHP4*, *StRR1*) homeodomains (*WUSCHEL*, *POTH1*, *BEL5*), auxin signaling, *ARF5*, carbon metabolism (*TPI*, *TIM*), protein synthesis, *NAC5* and a morphogenetic regulator of tuberization (*POTH15*).

Differential gene expression was found between TCS and cytokinin signaling genes (*HK1*, *HP4*), auxin signaling gene (*ARF5*), a homeodomain (*WUSCHEL*), carbon metabolism (*TPI*, *TIM*), protein synthesis (*NAC5*) and master regulators of tuberization (*BEL5*, *POTH1*, *POTH15*) during microtuber development.

According to our results derived from gene expression analysis and the network developed in STRING database, the molecular mechanisms can be interpreted as follows: 1.—Histidine-kinase1 (*StHK1*) is part of the two-component signaling system and was upregulated in MR8-G6-2iP (osmotic stress) treatment during the first two weeks of microtuber induction. This finding correlates with microtuber development in the first two weeks. Downregulation of *StHK1* occurred when potato microtubers were already formed, in the third and fourth week of induction. *AtHK1* is induced by cytokinins [73], functions as an osmosensor, a positive regulator of drought in Arabidopsis [74,75,76], salt stress [75,76], ABA signaling [77,78,79,80,81], ethylene [82,83] and stomatal control [84], and is related to MAPK cascade signaling [74] as well as conferring drought tolerance, by regulating levels of ABA accumulation [78,79,80,81]. *AtHK1* is required in the regulation of desiccation process during seed formation. *AHK1* mutation in Arabidopsis causes a decrease in storage proteins [83]. 2.—*AHK1* interact and feeds phosphate to phosphorelay-integrating histidine protein AHP4; in our analysis, it was upregulated in osmotic stress (MR8-G6-2iP) treatment in the second and third week of incubation. Singh et al. [85] found that *AHP4* was upregulated in root tissue of Arabidopsis under osmotic stress whereas other members of this family (*AHP1*, *AHP2* and *AHP3*) were downregulated. 3.—Histidine kinase 3 (*StHK3*) interacts with *AHP4*. It was found downregulated in three consecutive weeks (1 to 3) in osmotic stress condition MR8-G6-2iP and upregulated in the fourth week of incubation. *AHK3* is a negative regulator of the adaptive response to osmotic stress in Arabidopsis [75]. 4.—The response regulator type B, *StRR1*, was upregulated in the fourth week of incubation. In the two-component signaling system, phosphate is transferred to an aspartate residue within the receptor domain of the response regulator and this functions as a transcription factor. *StRR1* is involved in DNA repair, oxidative stress response, drought tolerance and low temperatures [76,77,86,87,88], and as a transcriptional activator of *WUSCHEL* [89] through a cytokinin-dependent signaling, which demonstrates its relevance and confirms its role in the tuberization signaling system. Lomin et al. [52] verified the rapid activation of type A response regulators in the presence of cytokinins in potatoes. 5.—Following the network, *WUSCHEL* interacts with *AHP4*. In our analysis, *WUSCHEL* was upregulated in the first and second week of incubation and downregulated from third to fourth week of incubation. *WUSCHEL* plays a central role during the specialization of stem cells in meristems [60,90,91]. Upregulation of *WUS* allows us to infer that this is essential for the initiation of microtuberization and its downward regulation can promote microtubers’ dormancy. 6.—*WUSCHEL* interacts with *POTH1*. It was found upregulated in the first and second week of incubation similar than *WUSCHEL* in MR8-G6-2iP medium. This indicates that *POTH1* is important in the regulation of the initiation and maturation of tubers. In potato, overexpression of *POTH1* results in improved tuberization rate in light and dark conditions [92]. *POTH1* has been largely analyzed during tuber development. When overexpressed along with *BEL5*, it increases cytokinin levels at the tips of stolons and promotes tuberization [44]. In addition, their homeodomain *KNOX* allows them to form dimers for the suppression of other genes [93]. Interaction of *POTH1* protein with seven members of the BEL family has been analyzed in the potato plant. Using mutant analysis, multiple BEL proteins that bind *POTH1* were identified, which implies that they are involved in a complex development control system in potato [44]. 7.—Next in the network, *POTH1* interacts with *BEL5*, a bell-like homeodomain protein 5, that was upregulated only in the second week of incubation. *BEL5* has the POX homeodomain, which allows it to form heterodimers with *POTH1* and these dimers are determining factors in the maintenance of the inflorescence meristem in Arabidopsis [94]. *BEL5* is a mobile long-range signaling molecule that promotes tuberization in potato [65]. *BEL5* is induced by light and has a primary role in tuber maturation [95]. 8.—*WUSCHEL* also interacts with *POTH15*, an ortholog of *STM* in Arabidopsis. In our analysis, *POTH15* was upregulated in the second and fourth week of incubation. *POTH15 (STM)* mutant causes defects in the formation and maintenance of SAM [96]. According to Endrizzi et al. [97], *STM* regulates the expression of *WUS*. Scofield et al. [96] showed that *STM* prevents the meristem organizing center cells by adopting specific lateral destinations. *WUSCHEL* and *STM* gene regulation converge in suppression of differentiation, coexpression of both genes produces a synergistic effect [98]. 9.—*ARF5*, an auxin response factor 5, interacts with *AHP4*, *POTH15*, *WUSCHEL*, *CLAVATA1* and *PIN1. ARF5* was upregulated in the second and fourth week of incubation in MR8-G6-2iP (osmotic stress) medium. *ARF5* mediate organ formation and vascular tissues throughout the Arabidopsis life cycle, and its expression is gradually restricted to the vasculature as the organs mature [99,100]. Upregulation in the second week is coincident with tuber enlargement. 10.—Alfa-nascent chain-associated (*NAC5*) complex is a highly conserved protein complex and was upregulated in the second and third week of incubation. *αNAC* plays an important role in conjunction with its heterodimer *βNAC* as regulator and as a chaperone in protein synthesis during translation, joining newly synthesized polypeptides to protect them from proteolysis and facilitate their folding in ribosome biogenesis and protein assembly; and in transport to other organelles, there is evidence that translation is less intense when *NAC* is absent from the ribosome [101]. The substantial increase in *αNAC* expression at the second week is attributed to cell division and growth in tuber formation. In the network, *αNAC* interacts with S10 ribosomal protein, that interacts with *THY-1* and thereby indirectly to the isoforms *TIM* and *TPI*. Furthermore, *THY-1* interacts with two histidine-kinase proteins, *HK1*, part of a two-component signaling system and *HK4 (WOL/CRE1)* which is a primary cytokinin receptor [102]. 11.—In the interaction network, triosephosphate isomerase (*TPI* and *TIM*) interacts with the bifunctional dihydrofolate reductase-thymidylate synthase, *THY-1,* involved in the de novo biosynthesis of dTMP nucleotide, and metabolism of folate [103]. *TIM* and *TPI* were found expressed opposed. *TIM* was upregulated in the first three weeks and *TPI* downregulated. Both *TPI* and *TIM* play crucial roles in glycolysis and glycogenesis [104]. *TIM* was found to be upregulated in a proteomic analysis of rice roots under hydric stress, due mainly to the supply of additional energy in form of ATP required to maintain homeostasis under these conditions [105]. Similar regulation pattern under hydric stress conditions was reported in maize (cytosolic) [106] and rice leaves (cytosolic) [107]. Upregulation during induction, initiation and enlargement of the tuber in the transcriptional analysis of *TIM*, and the primary interactions of *TIM* and *TPI* in the network, indicate its importance to carry out the primary mechanisms of tuber induction, initiation and enlargement, and the enzymes are essential in carbohydrate metabolism and energy generation.

Further analysis of each component is required to validate this network. Transcriptomic analysis, overexpression and downregulation of these genes are currently in evaluation.

## 4. Materials and Methods

### 4.1. Plant Material

#### 4.1.1. Potato Shoot Micropropagation

Potato cv. Alpha plantlets were propagated by shoot proliferation under in vitro conditions in MS medium [62], supplemented with 30 g/L sucrose (CAT 57-50-1 Sigma-Aldrich, St. Louis, MO, USA), activated charcoal (CAT: 242276 Sigma-Aldrich, St. Louis, MO, USA) (0.3%), pH 5.8 and solidified with 3 g/L gelrite (GELZAN CAT. G1910 Sigma-Aldrich, St. Louis, MO, USA). Shoots were incubated at 25/17 °C under fluorescent light at 25 µmol/m^2^/s of irradiance.

#### 4.1.2. Potato Microtuber Induction

Potato microtuber induction was induced by culture of stem segments containing two internodes and leaves, about 3 cm length, in flasks containing MR8-G6-2iP medium: MS medium, supplemented with 2iP 10 mg/L, 8 % sucrose, 6 g/L gelrite, activated charcoal (0.3%), pH 5.8. As control medium, MR1-G3-2iP (MS medium supplemented with 1% sucrose, 3 g/L gelrite and 10 mg/L 2iP) was used. Containers were sealed with plastic and incubated in dark at 25/17 °C for 31 dayss of incubation.

#### 4.1.3. Isolation of RNA and qPCR Analysis

Total RNA derived from four times (8, 15, 23, 31 dayss) was isolated using Trizol (Invitrogen, Carlsbad, CA, USA), RNA concentration was measured by its absorbance at 260 nm, ratio 260 nm/280 nm was assessed, and its integrity confirmed by electrophoresis in agarose 1% (*w*/*v*) gels. Samples of cDNA were amplified by PCR using SYBR^TM^ Green (ThermoFisher CAT: 4312704, Waltham, MA, USA) in Real-Time PCR Systems (CFX96 BioRad, Herules, CA, USA). The expression of *EF1* and *SEC3* was used as reference for calculating the relative amount of target gene expression using the ^2−ΔΔ^CT method [108,109]. qPCR analysis was based on at least five biological replicates for each sample with three technical replicates.

#### 4.1.4. Interaction Analysis of Proteins Directly Involved in Tuberization

A gene network with a confidence (0.500) was performed with STRING [110]; based on *S. tuberosum*, homologous genes present in *S. tuberosum* genome in Sol genomics network [111]. The selected genes were involved in cytokinin signaling, *AHK1*, *AHK3*, *AHP4*, *ARR1**,* auxin signaling; ARF5 (monopteros), homeodomains; *WUSCHEL*, *BEL5*, *POTH1*; master regulator *POTH15*, carbon metabolism *TIM*, *TPI* and protein folding *NAC5*. Gene identifier (Id) was made according to UNIPROT [112], NCBI [113] database. Homologous in *S. tuberosum* greater than 60% in protein sequence with *A. thaliana* were considered. Oligonucleotides were designed to qPCR (2^−^^ΔΔCT^ method analysis) gene expression or transcriptional analysis (Table 1).

## 5. Conclusions

Microtubers were suitable plant material for the analysis of gene expression.High content of sucrose (8%) and gelrite (6 g/L) enhances microtuber number, size and germination compared to non-osmotic medium.Differential gene expression of genes analyzed in microtubers induced under osmotic stress confirmed the hypothesis that TCS and cytokinin signaling are coupled with genes that have been associated in tuberization.Improvement of the understanding in molecular mechanisms involved in potato microtuberization was achieved by STRING database bioinformatic tool.

## Figures and Tables

**Figure 1 plants-10-00876-f001:**
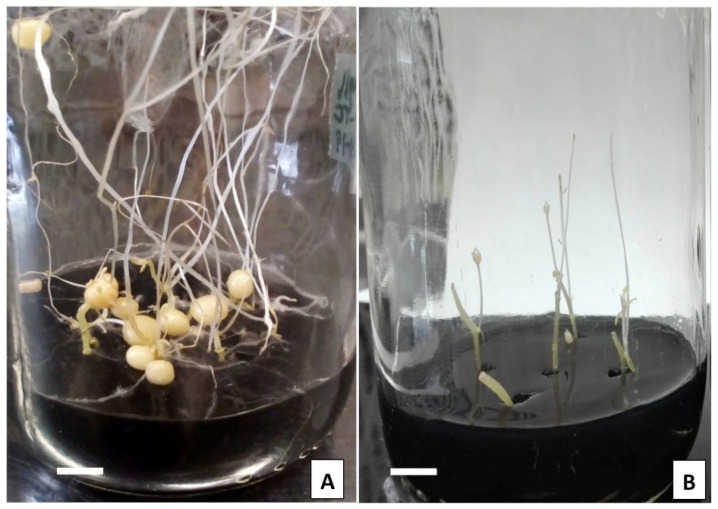
(**A**). Microtuber development of potato *S. tuberosum* cv. Alpha in medium with 8% sucrose, 6 g/L gelrite and activated charcoal after four weeks of incubation in darkness. (**B**). Shoot explants of potato in medium with low content of sucrose (1%), gelrite 3 g/L and activated charcoal after four weeks of incubation in darkness. Bar represents 1 cm.

**Figure 2 plants-10-00876-f002:**
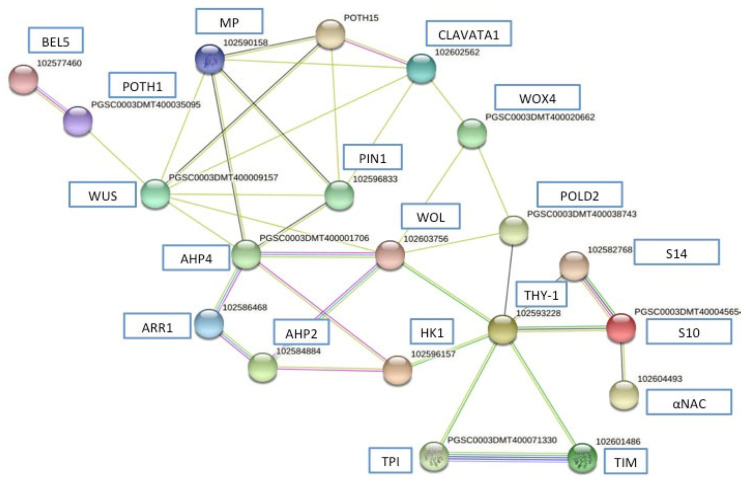
Gene network derived from STRING database of potato *S. tuberosum* (0.500 confidence) analyzed using qPCR. Homologous sequences of *A. thaliana* are written in rows. Numbers in the bubble indicate the gene identifier of each gene derived from the potato genome. The figure represents a full network, the edges indicate both functional and physical protein associations.

**Figure 3 plants-10-00876-f003:**
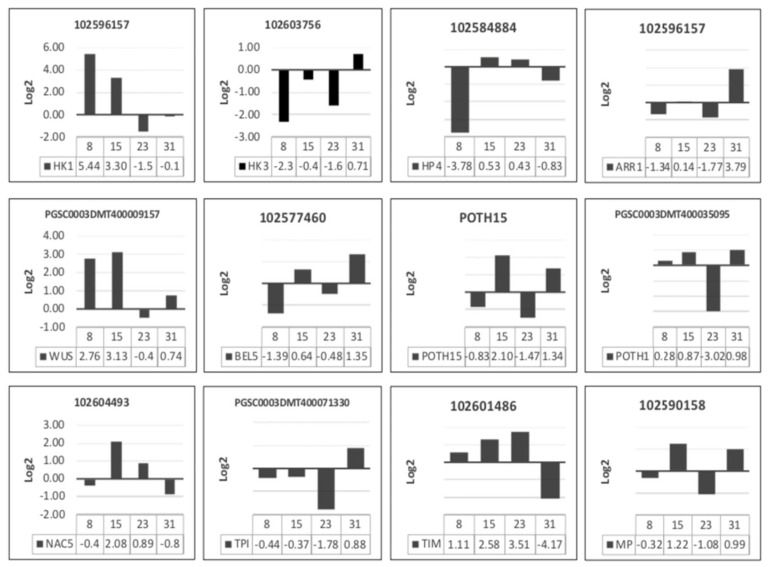
Gene expression analysis by qPCR during microtuber induction in osmotic stress medium (MR8-G6-2iP) after 8, 15, 23 and 31 dayss of incubation in darkness. Control medium was non-osmotic medium (MR1-G3-2iP). Relative expression levels plotted based on Log2. Normalized with *EF1* α (elongation factor 1 α) and *SEC3* (exocyst complex component 3).

**Figure 4 plants-10-00876-f004:**
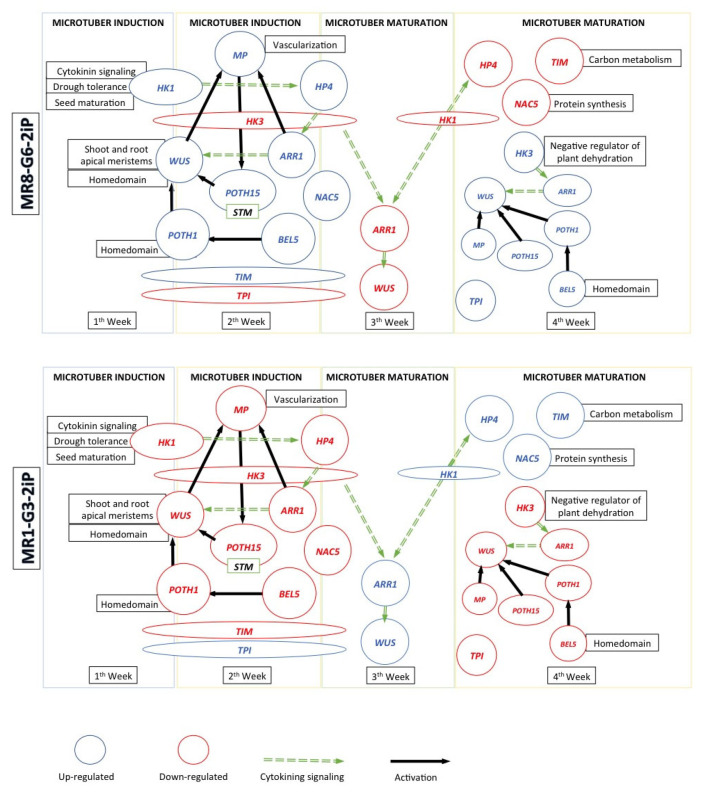
A model of microtuberization regulatory network in two conditions; osmotic stress (MR8-G6-2iP) and non-osmotic stress (MR1-G3-2iP).

**Table 1 plants-10-00876-t001:** Primer design of genes that were analyzed during microtuber induction of potato *S. tuberosum* cv. Alpha.

Gen ID	Sequences	Tm (°C)	%GC	Amplicon	ID NCBI
*StHK1*	L: TCGGAAAGCTCTGAATTCGT R: ATTCCATCCTTGACGAGACG	64.4 64.3	45% 50%	169 pb	XM_006340693.2
*StHK3*	L: GTTCATGCAGGTTGGTCCTT R: TCATGCCATGGAAATCTGAA	64.5 64.5	50% 40%	242 pb	XM_006352114.2
*StRR1*	L: TGTTGGGTCAGTGGTGAAAA R: CTCCGCTCGATTAGACTTGG	64.2 64	45% 55%	199 pb	XM_006345914.1
*StHP4*	L: AAACGCCAAGCTGCTTACAT R: TAGGGTCCATTTTCCAATGC	64 63.9	45% 45%	188 pb	XM_015315420.1
*StWUS*	L: TTTCACATGGGTTGGTGTTG R: CTTCATGCATGGGGAAAAGT	64.2 64.8	45% 45%	180 pb	XM_006340669.2
*StBEL5*	L: AACGCGAAAAAGCAAAGAAA R: GAAAATTCGCGGTCATTTGT	63.9 64	55% 55%	187 pb	NM_001287992.1
*POTH15*	L: CTCGTCTCTTGGCTGCTTATC R: CTACTACTGTTACGGCCCATTG	62.6 64.3	52% 50%	115 pb	XM_006348098.2
*POTH1*	L: AGAGACGATTACGCGGATAAAG R: GTCAAGATGGACGAGTTGGTATAG	63.5 62.9	45% 45%	101 pb	NM_001288425.1
*NAC5*	L: GAGCAGGAGCGAGAAGAAGA R: TCCTCAATCTTTGCCTCACC	64.3 64.5	55% 50%	183 pb	XM_006340649.2
*TIM*	L: GCCTGTTTGGGCTATTGGTA R: TCAGCAGCCTTGATGATGATGTC	64 64	50% 50%	249 pb	CP055237.1
*TPI*	L: CGGTGAACCAAACACATTGA R: GAGAAGCTAAAGAAGTTGCCATT	64.9 62	45% 39%	153 pb	NM_001318582.1
*MP*	L: TGGAAAGATGGGTTCTTTGG R: CCTGCATTCCCTTCAACAAT	64.1 64.2	45% 45%	157 pb	XM_006341964.2
*SEC3*	L: GATCTGCGGAAGGTGGTAAA R: CAGCAACTCCTCTGAGGTTAAG	62.3 64.3	57% 58%	102 pb	XM_006342542.2
*Ef1-alpha*	L: GCACTGGAGCATATCCGTTT R: TTTGGCCCTACTGGTTTGAC	64.2 64.1	58% 58%	244 pb	NM_001288491.1

## Data Availability

Data is contained within the article.

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
