# Peer review of "Gene Expression Analysis of Microtubers of Potato *Solanum tuberosum* L. Induced in Cytokinin Containing Medium and Osmotic Stress"

_plants, 2021, doi:10.3390/plants10050876_

Round 1
Reviewer 1 Report
General comments:
- Methodology: The authors reduced the concentration of sucrose and gelrite in the “non-osmotic” culture medium MR1-G3-2iP (as a control). This medium cannot be the control for osmotic stress treatment, since two factors (components) differ. In addition, by reducing the concentration of sucrose, they made a “non-inductive” culture media for potato tuberisation. According, to Xu et al. (1998) [Xu X, Vreugdenhil D, van Lammeren AAM (1998) Cell division and cell enlargement during potato tuber formation. J Exp Bot 49:573–582], high concentration of sucrose in the culture medium may decrease GAS concentration in the stolon and favours tuber formation. In my opinion, the differences between the culture media were not only due to the different osmotic pressure.
In fact, the authors studied the gene expression in an “inductive” and in a “non-inductive” for tuberisation culture medium. Thus, changes in gene expression may be attributed to other factors that favour tuberisation, such as hormones ratio (e.g. abscisic acid, gibberellins, ethylene), enzyme activities (e.g. sucrose synthase, invertases). For this reason, I suggest the authors to rewrite the title of the study, the abstract, the discussion section and the conclusions. I strongly encourage the authors to discuss the results of their work considering the literature on the physiology of tuberisation in potato (in vitro and in vivo).
- Literature concerning the physiology of tuberisation in potato is missing.
- In the keywords, prefer to use words that are not in the title.
- Please use italics for “in vitro”, “Arabidopsis thaliana”, “Pseudomonas elodea”.
- Please use upper words in the first letter of the name of the cultivar “alpha”.
Specific comments
Introduction
P1.L40: please replace the word “tuberization” with the word “tubers”
P2.L51 – 52: please replace the word “its” with the word “their”
Results
P3.L95 (Figure 3): Please write in the legend of the figure what does each of the images show.
P5.L160-164 (Figure 4): Please write numbers or letters in the images of the figure and write in the legend of the figure what does each of the images show.
Discussion
P6.L178: Please replace the word “giberelins” with the word “gibberellins”.
Materials and methods
Please provide details about the materials (gerlite, sucrose, cytokinin) you used (e.g. trademark).
References
The references section is not written according to the instructions to authors. The authors must make corrections carefully.
For example:
Use dot only in abbreviations of the journal title (P9.L298: replace “Physiol Plantarum.” with “Physiol. Plantarum”, …)
Some abbreviations are not correct (P9.L329, P9.L331, P9.L332……….)
Use abbreviations of the journal title (P10.L400, ….. P11.L405)
Use italics for journal title (P10.L396)
The names of the authors are not written according to the instructions (e.g. do not use “&”).
The year of publication is not written in the right place (P9.L330, P9.L336, P9.339.…)
Author Response
Open Review
(x) I would not like to sign my review report
( ) I would like to sign my review report
English language and style
( ) Extensive editing of English language and style required
( ) Moderate English changes required
(x) English language and style are fine/minor spell check required
( ) I don't feel qualified to judge about the English language and style
|
|
Yes |
Can be improved |
Must be improved |
Not applicable |
|
Does the introduction provide sufficient background and include all relevant references? |
( ) |
(x) |
( ) |
( ) |
|
Is the research design appropriate? |
( ) |
( ) |
(x) |
( ) |
|
Are the methods adequately described? |
( ) |
( ) |
(x) |
( ) |
|
Are the results clearly presented? |
( ) |
(x) |
( ) |
( ) |
|
Are the conclusions supported by the results? |
( ) |
( ) |
(x) |
( ) |
Comments and Suggestions for Authors
General comments:
- Methodology: The authors reduced the concentration of sucrose and gelrite in the “non-osmotic” culture medium MR1-G3-2iP (as a control). This medium cannot be the control for osmotic stress treatment, since two factors (components) differ. In addition, by reducing the concentration of sucrose, they made a “non-inductive” culture media for potato tuberisation.
Dear Reviewer 1:
In this work we analyzed during microtuberization of potato the relative expresión of genes that have been previously investigated as master regulators of potato tuberization (BELL5, POTH1, POTH15) and genes involved in the two component system and cytokinin signaling as well as genes invoved in carbon metabolismo and protein syntesis.
In preliminary experiments we found that a medium with 8% sucrose coupled with high content of gelrite and 2iP as cytokinin, that we called (osmotic stress medium) was the best medium to induce a higher amount of microtubers and in shorter time compared with only sucrose 8%, sucrose 8% plus gelrite 6 g/L.
As you may know, in “some “ scientific papers have claimed that 8% sucrose is not a osmotic stress compund during potato microtuberization.
But, by definition: Osmotic stress is a sudden change in solute(s) concentration(s) around a cell, causing a rapid change in the movement of water across its cell membrane. Under conditions of high concentrations of either salts, substrates or any solute in the supernatant, water is drawn out of the cells through osmosis. The solute potential of a 0.1 M solution of distilled water and sucrose at 20 ºC at standard atmospheric pressure is - 0.23. If we continue adding sucrose to the solution until it reaches a concentration of 0.75 M at 20 ºC at standard atmospheric pressure, the solute potential continues to drop to a value of -1.87.
This information was taken from: Ford, R. 2015. Demonstrating Osmotic Potential: One Factor in the Plant Water Potential Equation. Article 29 in Tested Studies for Laboratory Teaching, Volume 36 (K. McMahon, Editor). Proceedings of the 36th Conference of the Association for Biology Laboratory Education (ABLE). http://www.ableweb.org/volumes/vol-36/?art=29.
Taking in consideration this basic information, sucrose at 8% act as osmotic stress inducer, and what molecular mechanisms ocurred with the use of high content of sucrose during microtuberization is a scientific question that we intend to answer.
Sucrose has been found to be involved in several molecular mechanisms:
1.- Sucrose metabolism
2.- In storage proteins
3.- In development
4.- In floral induction
5.- Sucrose signaling
Several genes have been shown to be affected by sucrose, among others;
1.- WRKY20, WRKY family transcription factor family protein; Transcription factor. Interacts specifically with the W box (5'-(T)TGAC[CT]-3'), a frequently occurring elicitor- responsive cis-acting element (By similarity) (557 aa)
2.- SUT4, Sucrose transport protein SUC4; Responsible for the transport of sucrose into the cell, with the concomitant uptake of protons (symport system). Can also transport maltose at a lesser rate. May also transport biotin; Belongs to the glycoside-pentoside-
3.- ATHXK4 Hexokinase-4; Fructose and glucose phosphorylating enzyme. May be involved in the phosphorylation of glucose during the export from mitochondrion to cytosol (By similarity) (502 aa)
If we analyzed this three sucrose-activated genes in a bioinformatic tool STRING database together with the genes we analyzed in this work you will find that they interact and the molecular, biochemical and physiological functions are related with abiotic stimulus, water deprivation, osmotic stress, water stress, abiotic stress, hormone signaling pathways (auxin, cytokinin, gibberelins, etc.).
- This medium cannot be the control for osmotic stress treatment, since two factors (components) differ. In addition, by reducing the concentration of sucrose, they made a “non-inductive” culture media for potato tuberisation.
Sucrose and gelrite act a osmotic stress inducer, you have to compare it with the opposite concentration, low amount either sucrose or gelrite. In order to get microtubers of larger sized the media should be supplemented with cytokinin and higher sucrose level. It is thought that the sucrose dissociates to allow a higher osmotic potential within the cells. Thus, the role of sucrose in plant tissue culture media as an osmoticum as well as a carbohydrate source has been established.
- According, to Xu et al. (1998) [Xu X, Vreugdenhil D, van Lammeren AAM (1998) Cell division and cell enlargement during potato tuber formation. J Exp Bot 49:573–582], high concentration of sucrose in the culture medium may decrease GAS concentration in the stolon and favours tuber formation.
No answer
- In my opinion, the differences between the culture media were not only due to the different osmotic pressure.
It was answered
In fact, the authors studied the gene expression in an “inductive” and in a “non-inductive” for tuberisation culture medium.
Thus, changes in gene expression may be attributed to other factors that favour tuberisation, such as hormones ratio (e.g. abscisic acid, gibberellins, ethylene), enzyme activities (e.g. sucrose synthase, invertases).
No answer
For this reason, I suggest the authors to rewrite the title of the study, the abstract, the discussion section and the conclusions.
No answer
I strongly encourage the authors to discuss the results of their work considering the literature on the physiology of tuberisation in potato (in vitro and in vivo).
- Literature concerning the physiology of tuberisation in potato is missing.
- In the keywords, prefer to use words that are not in the title.
- Please use italics for “in vitro”, “Arabidopsis thaliana”, “Pseudomonas elodea”.
- Please use upper words in the first letter of the name of the cultivar “alpha”.
No answer

Reviewer 2 Report
The authors have investigated the molecular mechanisms of microtuber development in potato using a rapid and efficient protocol based on osmotic stress and cytokinin signaling. They conducted gene expression analysis of genes involved in multiple metabolic pathways using qPCR real time to have identified the differentially expressed genes during microtuber development.
I believe that this manuscript contains important scientific merits that are worth of being published. However, I strongly believe that this manuscript is not ready for publication due to its poorly organized presentation throughout the entire manuscript. I will list some but not all of these problems here:
For example, I have seen a lot of broken sentences, e.g., Lines 35-37, 41 and 42, and 177-179, throughout the entire manuscript. I really mean the entire manuscript, if the authors only revise these three listed here, then it is not sufficient at all. The authors should really pay attention to the presentation of the information and try to improve the English and devote serious efforts to correcting those numerous grammatical and editorial errors.
Figure 4, please simplify the unnecessary coloring on this figure unless if the authors believe that the coloring is required to deliver the information presented on this figure.
Lines 220-222, I believe it is necessary to present these molecular mechanisms in the text of the manuscript but not in the supplementary materials.
Lines 223-226, the authors indicated that differential gene expression was found…so what? I believe that the authors should continue to complete this discussion of this discovery.
Author Response
Comments and Suggestions for Authors
The authors have investigated the molecular mechanisms of microtuber development in potato using a rapid and efficient protocol based on osmotic stress and cytokinin signaling. They conducted gene expression analysis of genes involved in multiple metabolic pathways using qPCR real time to have identified the differentially expressed genes during microtuber development.
I believe that this manuscript contains important scientific merits that are worth of being published.
However, I strongly believe that this manuscript is not ready for publication due to its poorly organized presentation throughout the entire manuscript. I will list some but not all of these problems here:
For example, I have seen a lot of broken sentences, e.g., Lines 35-37,
It was corrected.
A potato tuber is a specialized stem that arises from the underground organ known as stolon [3]. Tubers are used for plants survival by vegetative propagation, they are sink organs in which surplus photosynthetic assimilates are stored [4], starch [5,6], vitamins [7] and proteins [8,9] as the main storage components.
41 and 42,
It was corrected.
The induction of microtubers under in vitro conditions was first described by Baker [14]; Mes and Menge [15] as plant pathology tool. An extensive number of articles since then, have been published for basic and practical research (For review; [16-19]).
and 177-179, throughout the entire manuscript.
It was corrected.
In plant tissue culture medium with 2-3% sucrose lacking plant growth regulators, microtubers can be induced after 4-5 months in culture. However, microtubers are significanty accelerated and improved by using plant growth regulators or by changing culture conditions.
An extensive number of publications about potato microtuber induction using plant growth regulators has been made; auxins [50], gibberellins [51 - 55], strigolactones [56], abscisic acid [57], ethylene [58], jasmonate [3, 59 - 62].
I really mean the entire manuscript, if the authors only revise these three listed here, then it is not sufficient at all.
The authors should really pay attention to the presentation of the information and try to improve the English and devote serious efforts to correcting those numerous grammatical and editorial errors.
The manuscript has been changed, you can find in the corrected mansucript with highlighted blue and yellow colour.
Figure 4, please simplify the unnecessary coloring on this figure unless if the authors believe that the coloring is required to deliver the information presented on this figure.
It was corrected. Colored figure was changed
Lines 220-222, I believe it is necessary to present these molecular mechanisms in the text of the manuscript but not in the supplementary materials.
It was corrected. The molecular mechanisms were included in the discussion
Lines 223-226, the authors indicated that differential gene expression was found…so what? I believe that the authors should continue to complete this discussion of this discovery.
It was corrected.

Reviewer 3 Report
The manuscript of paper by Herrera-Isidron L. et al. “Gene expression analysis of microtubers of potato Solanum tuberosum L. induced in cytokinin containing medium and osmotic stress” provides new data on the expression of developmental and plant hormone related genes during microtuber formation. Using STRING database authors supposed gene network acting during microtuberisation of potato based on obtained qPCR data. The manuscript contains lots of errors in formatting and design of text, figures as well as in English.
1) Potato is the fourth-fifth most important crop worldwide after maize, sugarcane, rice/wheat (www.fao.org). At least it is not third one (as is written at line 33).
2) Authors not always use correct Italic formatting for plant species and genes’ names throughout the text (lines 24, 29, 53, 55, 60, 104, 105, 115, 187, 192, 200, 216-219, 259) as well as in the legends for Figures 2-4 and in Table S1. On the contrary line 24 of supplementary contains excessive Italic formatting.
3) There are a lot of inaccuracies in the text as well. Line 15: word techniques misses “t”. Line 16: i.e. misses dots. Sentences at lines 49-50 and 57-59 are inconsistent; there is no predicate or noun. Line 241 possesses excessive spacebar. Lines 236, 252 show incorrect superscript/subscript formatting.
4) The name of cultivar usually starts with uppercase letters (lines 81, 89).
5) Figure 2 must be improved. Gene names have to be formatted in Italic and totally match to the text. What do numbers on the Figure mean? Maybe it is better to change it to A, B, C, etc. and to explain in the Figure caption?
6) Text S1 from supplement is better to transfer into “Discussion” section, since this section lacks discussion itself. Maybe all materials form supplement is better to put in correspondent sections of the paper (primer’s table into methods)?
Taking all mentioned above into account, the manuscript would excellently merit publishing in Plants, but it requires serious revision.
Author Response
1) Potato is the fourth-fifth most important crop worldwide after maize, sugarcane, rice/wheat (www.fao.org). At least it is not third one (as is written at line 33).
It was corrected. Potato (Solanum tuberosum L.) is the fourth
2) Authors not always use correct Italic formatting for plant species and genes’ names throughout the text (lines 24, 29, 53, 55, 60, 104, 105, 115, 187, 192, 200, 216-219, 259) as well as in the legends for Figures 2-4 and in Table S1. On the contrary line 24 of supplementary contains excessive Italic formatting.
It was corrected lines 24-29, 53, 55, 60, 104, 105, 115, 187, 192, 200, 216-219
3) There are a lot of inaccuracies in the text as well. Line 15: word techniques misses “t”.
It was corrected.
Abstract: Potato microtuber production through in vitro techniques are ideal
Line 16: i.e. misses dots.
It was corrected. Microtuber development is influenced by several factors, i.e. high content sucrose and cytokinins
Sentences at lines 49-50
It was changed. It has been well documented the effect of cytokinins (CKs) during microtuber development, vascularity and root induction of potato [21 - 27].
and 57-59 are inconsistent; there is no predicate or noun.
It was changed. In addition, CKs participate in plant defense against biotic and abiotic adverse factors [30 - 32].
Line 241 possesses excessive spacebar.
It was corrected.
Lines 236, 252 show incorrect superscript/subscript formatting.
It was corrected, 25 mmol/m2 /sec
4) The name of cultivar usually starts with uppercase letters (lines 81, 89).
It was corrected.
5) Figure 2 must be improved.
Gene names have to be formatted in Italic and totally match to the text.
These are protein names, they are not italized.
What do numbers on the Figure mean?
Numbers indicate de gene number in the potato genome: Solanum tuberosum. In this case we use STRING database:
https://stringdb.org/cgi/input?sessionId=boX8sUCqktWK&input_page_active_form=single_identifier
i.e. HK1 in potato is 102596157. The information in this database is:
Uncharacterized protein; Cold inducible histidine kinase 1 Identifier: PGSC0003DMT400009219, 102596157 Organism: Solanum tuberosum
In the figure network, we add the correspondence protein name given in Arabidopsis thaliana. Protein names are not italized.
Maybe it is better to change it to A, B, C, etc. and to explain in the Figure caption?
We decided to use the identifier gene of potato as evidence in the case of someone would like to check the network using string or another bioinformatic tool. Adding letters would be like a model. On the contrary using this strategy, you can analyze the information written in this manuscript by yourself and do better mining database analyzis.
6) Text S1 from supplement is better to transfer into “Discussion” section, since this section lacks discussion itself.
It was corrected. We have transferred the supplementary text to the discussion.
Maybe all materials form supplement is better to put in correspondent sections of the paper (primer’s table into methods)?
It was corrected. We have transferred the supplementary table to materials and methods.
Taking all mentioned above into account, the manuscript would excellently merit publishing in Plants, but it requires serious revision.
Submission Date
06 March 2021
Date of this review
22 Mar 2021 21:30:08
Round 2
Reviewer 1 Report
I appreciate the effort the authors have made to improve their manuscript.
The authors referred, in their response to reviewer’s comments, that “the role of sucrose in plant tissue culture media as an osmoticum as well as a carbohydrate source has been established”. However, the authors discuss the effect of the media with high gelrite and sucrose content considering only the osmotic stress induced by the media. I strongly encourage the authors to consider the effect of sucrose, as a carbohydrate source, on microtuberisation. They could also correct their conclusion (P11.L364), since the enhancement of tuberisation in the media with higher sucrose and gelrite content was not only due to the higher gelrite content, as they refer, but to the higher sucrose content (and as a carbohydrate source), as well.
Specific comments
P6.L181: Please add a reference.
P13.L458-459, P14.L524: The title of the manuscript is not written according to the instructions.
P13.L485: please use “GA3”
Some abbreviations of the journal title are incorrect: e.g. P14.L553, the abbreviation of Plant Physiology is Plant Physiol.
P14.L547, 555: T Plant Cell. is incorrect abbreviation
Author Response
Dear Reviewer 1, I have accepted your comments about this manuscript. You will find it within the manuscript.

Reviewer 2 Report
I appreciate very much the efforts that the authors have devoted to making improvement to their manuscript. I have no more questions.
Author Response
Dear Reviewer 2, I appreciate your time for reading and the comments you made of the manuscript.
